# Strong Genetic Effects on Bone Mineral Density in Multiple Locations with Two Different Techniques: Results from a Cross-Sectional Twin Study

**DOI:** 10.3390/medicina57030248

**Published:** 2021-03-08

**Authors:** Marton Piroska, David Laszlo Tarnoki, Helga Szabo, Zsofia Jokkel, Szilvia Meszaros, Csaba Horvath, Adam Domonkos Tarnoki

**Affiliations:** 1Medical Imaging Centre, Faculty of Medicine, Semmelweis University, 1082 Budapest, Hungary; piroska.marton@semmelweis-univ.hu (M.P.); tarnoki4@gmail.com (D.L.T.); szabo.helga.se@gmail.com (H.S.); zsofijokkel@gmail.com (Z.J.); 2Hungarian Twin Registry, 1082 Budapest, Hungary; 3Central Radiological Diagnostic Department, Medical Centre Hungarian Defence Forces, 1134 Budapest, Hungary; 4Department of Internal Medicine and Oncology, Faculty of Medicine, Semmelweis University, 1085 Budapest, Hungary; meszaros.szilvia@med.semmelweis-univ.hu (S.M.); horvath.csaba@med.semmelweis-univ.hu (C.H.)

**Keywords:** genetics, bone mineral density, lumbar spine, hip, quantitative bone ultrasound

## Abstract

*Background and Objectives*: Previous studies have demonstrated that risk of hip fracture is at least partly heritable. The aim of this study was to determine the magnitude of the genetic component of bone mineral density (BMD), using both X-ray and ultrasound assessment at multiple sites. *Materials and Methods:* 216 adult, healthy Hungarian twins (124 monozygotic, MZ, 92 dizygotic, DZ; mean age 54.2 ± 14.3 years), recruited from the Hungarian Twin Registry with no history of oncologic disease underwent cross-sectional BMD studies. We measured BMD, T- and Z-scores with dual energy X-ray absorptiometry (DEXA) at multiple sites (lumbar spine, femoral neck, total hip and radius). Quantitative bone ultrasound (QUS) was also performed, resulting in a calculated value of estimated bone mineral density (eBMD) in the heel bone. Heritability was calculated using the univariate ACE model. *Results:* Bone density had a strong genetic component at all sites with estimates of heritability ranging from 0.613 to 0.838 in the total sample. Lumbar BMD and calcaneus eBMD had major genetic components with estimates of 0.828 and 0.838 respectively, and least heritable (0.653) at the total hip. BMD of the radius had also a strong genetic component with an estimate of 0.806. No common environmental effect was found. The remaining variance was influenced by unique environment (0.162 to 0.387). In females only, slightly higher additive genetic estimates were found, especially in the case of the femoral neck and total hip. *Conclusion:* Bone mineral density is strongly heritable, especially in females at all locations using both DEXA and QUS, which may explain the importance of family history as a risk factor for bone fractures. Unshared environmental effects account for the rest of the variance with slight differences in magnitude across various bone regions, supporting the role of lifestyle in preventing osteoporotic fractures with various efficacy in different bone regions.

## 1. Introduction

Previous studies have demonstrated that risk of hip fracture is heritable. In a British classical twin study aiming to determine the genetic and environmental influence—using a classical twin study design by comparing monozygotic (MZ) and dizygotic (DZ) twins [1]. For important risk factors (such as postmenopausal bone mineral density (BMD), calcaneus and hip axis length), 500 healthy female twins (128 monozygotic, MZ and 122 dizygotic, DZ pairs, aged 50 to 70 years) underwent BMD measurement at multiple sites as well as hip axis length measurement and calcaneus ultrasound examination. A strong genetic component of BMD was found at all sites with the estimation of heritability ranging from 0.46 to 0.84. Furthermore, all the three risk factors proved to be independently associated with hip fracture and independently heritable [2].

BMD, as a major predictor of osteoporotic fractures, became a focus of attention with the growing number of genome-wide association studies (GWAS). Several common, uncommon and rare structural and copy-number variations were found in association with osteoporotic fractures, highlighting the importance of understanding its background more precisely [3,4,5,6,7,8,9]. However, identifying patients who will experience osteoporotic fracture from measurements of BMD are still difficult, and genetic versus environmental effects have not been assessed in multiple bone regions using both X-ray (dual energy X-ray absorptiometry, DEXA) and heel bone ultrasound technique. Since only females were involved in the previous U.K. twin study, the aim of this study was to determine the genetic component of BMD, using both X-ray and ultrasound assessment at multiple sites in both males and females.

## 2. Materials and Methods

### 2.1. Subjects and Study Design

A total of 216 adult, healthy Hungarian twins (124 MZ, 92 same-sex DZ; mean age 54.2 ± 14.3 years), recruited from the voluntary Hungarian Twin Registry with no history of oncologic disease underwent cross-sectional BMD studies in 2019 [10]. Pregnant subjects, patients with uncontrolled chronic cardiorespiratory disease (i.e., asthma exacerbation or acute heart failure) and those with an acute respiratory infection within 4 weeks of measurement were excluded. In the absence of genotyping and in order to maximize the accuracy of zygosity classification, we used a multiple-choice self-reported questionnaire. Zygosity was assigned according to a seven-part self-reported response [11]. Participants completed a questionnaire in order to identify clinical symptoms and to obtain complete past medical history and a list of medications. Smoking history was recorded as follows: each subject was categorized as never, former or active smoker. Pack-years were calculated as number of pack years = (number of cigarettes smoked per day × number of years smoked)/20. Weight was measured by OMRON BF500 monitor (Omron Healthcare Ltd., Kyoto, Japan). Body mass index (BMI) was determined by the weight (kg)/height (m)^2^.

The study was approved by the National Scientific and Ethics Committee (institutional review board number: ETT TUKEB 189-4/2014) and was carried out according to the principles stated in the Declaration of Helsinki. All subjects provided written informed consent.

### 2.2. Bone Mineral Density Assessment

Central (lumbar spine L1-L4, femoral neck and total hip) and peripheral (radius) bone DEXA scans (Hologic Horizon; Marlborough, MA, USA) were assessed. Heel bone quantitative ultrasonography (QUS) measurements were performed in all the subjects using the Sahara Clinical Sonometer (Hologic, Bedford, MA, USA). Ultrasound method does not measure BMD, but is derived from the measurement of quantitative ultrasound parameters (estimated BMD (eBMD)). The Sahara device measured both broadband ultrasound attenuation (BUA, dB/MHz) and speed of sound (SOS, m/s) at the calcaneus. The bone sonometer automatically estimated the heel bone mineral density (eBMD, g/cm^2^). T-score—the number of standard deviations (SD) that the absolute BMD is above or below the mean value for a healthy, same sex, young adult population—and Z-score—the number of SDs the absolute BMD is above or below the mean value for a healthy, age and sex matched population—were assessed. After consultation with a radiation protection expert, the amount of radiation was negligible compared to a transatlantic flight, about one=thousandth of the annual natural background radiation, and most of the radiation exposure came from low-energy X-ray photons. DEXA parameters were the following in case of lumbar spine: V = 76 kV, I = 3.0 mA, t = 0.31 min, D patient = 37 µGy, window: 15.1 cm × 14.0 cm, in case of radius: V = 76 kV, I = 0.15 mA, t = 0.21 min, D patient = 2 µGy, window: 10.4 cm × 24 cm, in case of total hip: V = 76 kV, I = 3.0 mA, t = 0.31 min, D patient = 37 µGy, window: 15.1 cm × 14.0 cm.

### 2.3. Statistical Analysis

Continuous variables are presented as means and standard deviation (SD), while categorical parameters are shown as numbers and percentages.Based on within-twin correlations between MZ and DZ twins (rMZ, rDZ), genetic structural equation models were built to quantify the proportion of genetic and environmental factors contributing to each parameter (ACE model) using OpenMX [12], a library within the R programming language [13] in order to break down the variance into additive genetic effects (A), common or shared (C), and unique or unshared (E) environmental effects [14]. Using the structural equations model, it is possible to decompose the variation between the twins assuming that MZ twins share nearly 100% of their genome, while DZ pairs share 50% on average. The A measures the effects due to genes at multiple loci or multiple alleles at one locus. The C estimates the contribution of the common family environment of both twins (familiar socialization, diet, early childhood, education in the same school, living in the same town, exposure to high levels of air pollution, shared womb, sharing similar socioeconomic status, etc.), whereas the unshared environmental component estimates the effects that apply only to each individual twin and includes measurement error. Confidence intervals were calculated using likelihood-based confidence intervals [15]. Submodels of the full ACE models were calculated to determine the most parsimonious model capable of correctly describing our data. If dropping one of the sources of variation (resulting in: CE or AE models) did not cause a significant deterioration in fit as compared to the full ACE model using a likelihood ratio test, the most parsimonious model with the best fit was selected based on Akaike Information Criteria. Calculations involving raw BMD values were corrected for age, sex and BMI, T-scores were adjusted for age and BMI, and Z-scores were only adjusted for BMI. A *p* value < 0.05 was considered statistically significant.

## 3. Results

### 3.1. Study Population

Baseline demographics and clinical characteristics of the overall population (N = 216) and by zygosity are summarized in Table 1. The mean age of the cohort was 54.2 ± 14.3 years (72% female) and patients in the DZ group were older and had higher radial BMD score than the MZ group (*p* < 0.05). Otherwise there were no significant differences among the groups.

### 3.2. Heritability of Bone Mineral Density

Based on comparing the base ACE model with the nested submodels using Akaike Information Criteria, the AE model showed the best model fit, meaning that the effects of common environmental factors were negligible. After seeing that the common environmental effects were negligible, we also fitted ADE models trying to differentiate between additive and dominant genetic effects; however, with this sample size we did not have enough power to reasonably differentiate between dominant and additive effects—the ADE and AE models did not differ significantly.

Bone density had a strong genetic component at all sites with estimates of heritability ranging from 0.613 to 0.838. Lumbar BMD and calcaneus eBMD had major genetic components with estimates of 0.828 and 0.838, respectively, and BMD was least heritable (0.653) at the total hip. Calcaneus eBMD T-score had also a strong genetic component with an estimate of 0.838. The remaining variance was influenced by unique environment (0.162 to 0.387). The results are shown in Figure 1 and in Table 2.

### 3.3. Differences between Genders

To examine if there are any differences in heritability between males and females, we reran the ACE models on the female part of our sample. Since we did not have enough male dizygotic pairs to repeat the calculations on an only-male sample, we compared the estimates of the female-only model with the estimates of the general model. The same methodology was applied on the female samples, which resulted in slightly higher additive genetic estimates, especially in the case of the femoral and total hip regions. (Figure 2). Only based on the estimates, a clear trend of elevated heritability in females is outlined in Figure 2, which may suggest that in females, the bone mineral density is more genetically predetermined.

## 4. Discussion

We demonstrated that bone mineral density had a strong genetic component at all sites measured with DEXA and ultrasound method as well, especially in females. Lumbar BMD and heel bone mineral density had major genetic components, and least heritable was at femoral neck and total hip. The remaining variance was influenced by unique environment, which was the lowest in lumbar spine and calcaneus, and highest in total hip and femoral neck.

Our findings are in line with results of other studies, which demonstrated BMD heritability of 0.6–0.8, meaning that 60–80% of the variation in BMD is inherited from parents and the remainder is derived from the environment [16,17]. Other studies investigated genetic effects of osteoporotic fracture as well, which is the endpoint clinical outcome of osteoporosis, with a heritability of 0.5–0.7 [18]. Comparing our results on both sexes with the one of the U.K. twin registry in 500 normal female postmenopausal twins [2], the heritability of lumbar BMD was similar (0.83 vs. 0.78), as well as of the hip (0.65 vs. 0.67), femoral neck (0.67 vs. 0.84), radius (0.81 vs. 0.61 in our sample and in the U.K. sample, respectively). Broadband ultrasound attenuation of the calcaneus had a moderate genetic component with an estimate of 0.53 in the U.K. sample, which was much higher, 0.838 in our sample [2]. The reason for the difference might be the different, older equipment used (McCue Cuba Clinical heel scanner) in the U.K. sample and the different study population (postmenopausal females with mean age of 60 years). Moreover, different parameters were measured with ultrasonography (BUA in the U.K., and eBMD in our sample). In this sense, the direct BUA measurement of the U.K. sample and our eBMD measurement derived partly from BUA but also from SOS do not really mean complete identity; the difference between the two types of data can also be explained by this methodological difference. Of note, ultrasound of the calcaneus reflects structural changes in bone reflecting both trabecular separation and connectivity [19]. Our data are consistent with these findings showing that there is a strong genetic component to both DEXA and calcaneal ultrasound-based BMD. Compared to the U.K. study, we reported higher heritability values in females only, especially in the case of the femoral neck. To further examine these differences and prove their significance will, however, require a larger sample size; this prompts further investigation. 

The two highest heritability values demonstrated in the heel bone and vertebrae were very close to each other, which is to be expected because both are dominated by trabecular bone stock, supporting the biological adequacy of the measurements. According to our study, genetic determination of the predominantly trabecular vertebrae and the predominantly cortical forearm is close to each other, whereas the mixed-composition hip bone is much lower. The highly load-bearing nature of the hip bone might explain why the role of the environment is greater in its development. A recent family study involving 177 mother–offspring pairs from 162 families demonstrated that genetic factors play an important role in the development of bone geometry, volumetric bone mineral density and microarchitecture of trabecular and cortical bone measured by high resolution peripheral quantitative computerized tomography at the distal radius and tibia [20]. This finding also explain our findings concerning the BMD heritability.

Albeit BMD and osteoporosis is highly heritable, the underlying genomic and molecular mechanisms are still largely unknown at an individual level. Genome-wide association studies identified hundreds of susceptibility loci, but no actionable genetic cause could be identified [17]. A recent study identified 28 variants of interest, but only 3 were classified as pathogenic or likely pathogenic variants: COL1A2 p.(Arg708Gln), WNT1 p.(Gly169Asp), and IDUA p.(His82Gln) [21]. A recent Chinese GWAS study reported that one SNP rs35282355 located in the human immunodeficiency virus type 1 enhancer-binding protein 3 gene (HIVEP3) and another 25 SNPs located in LINC RNA were associated with femoral neck BMD [22]. BMD and bone size, which is also an important factor contributing to osteoporotic fractures, were genetically correlated [23]. Omics technologies, such as transcriptomics, epigenomics, proteomics, metabolomics and metagenomics might provide further insights to the pathophysiology of osteoporosis, especially multiomics studies [17]. This knowledge could help to accurately identify patients who will experience osteoporotic fracture from measurements of BMD.

Beyond the high genetic effects, 16–39% of the variance was influenced by unique environment based on our findings. The environmental effect was the lowest in lumbar spine and calcaneus, and highest by total hip and femoral bones. Other studies reported that the nongenetic variance is attributed to the hip geometric parameters and tissue horizontal characteristics [17]. 

The BMD-associated loci identified so far in GWAS do not account for all the heritability in osteoporosis (“missing heritability”) due to the gene–environment interactions, such as smoking, diet and regular physical activity [24]. Moreover, recent studies revealed the role of gut microbiome in bone metabolism and health [25,26]. Future host and microbiome multiomics integration studies might lead to a major breakthrough in the prediction and therapeutic treatment of osteoporosis [17].

Our study has several limitations, including a relatively low sample size, which was reflected in the ACE modeling results of total hip BMD (with the lowest heritability estimate), where both AE and CE models could have been accepted based only on the likelihood ratio tests, which reflect a lack of power to unambiguously exclude one parameter.

## 5. Conclusions

In conclusion, our study suggests a strong role for heritability of BMD within an asymptomatic twin population at all locations using both DEXA and heel bone ultrasonography technique, especially in females. These findings may explain the importance of family history as a risk factor for bone fractures and might stimulate further studies in family risk-based osteoporosis screening due to the importance of detecting genetic risk factors and emphasizing the benefit of early diagnosis of osteoporosis. Unshared environmental effects account for the rest of the variance with slight differences in magnitude across various bone regions, supporting the role of lifestyle in preventing the adverse clinical outcomes associated with osteoporosis. Due to the differences of the magnitude of unique environments across various skeletal regions, further studies could investigate the bone-specific environmental and genetic effects to understand the individual differences in fracture regions in osteoporotic patients.

## Figures and Tables

**Figure 1 medicina-57-00248-f001:**
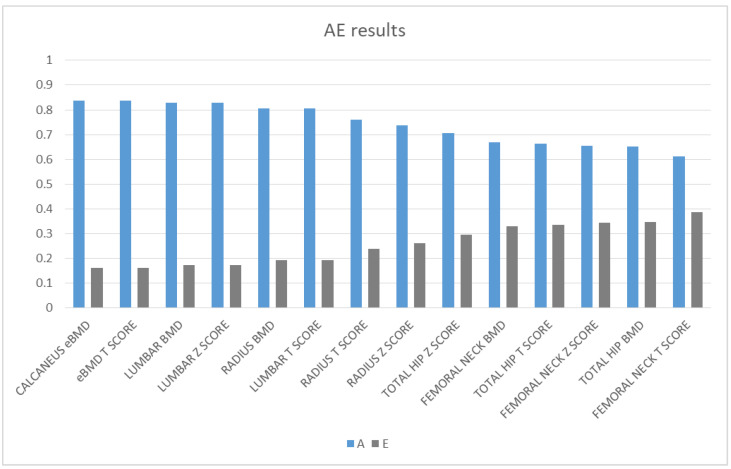
Magnitude of additive genetic (A) and unique environmental (E) factors of the investigated bone mineral density (BMD) phenotypes.

**Figure 2 medicina-57-00248-f002:**
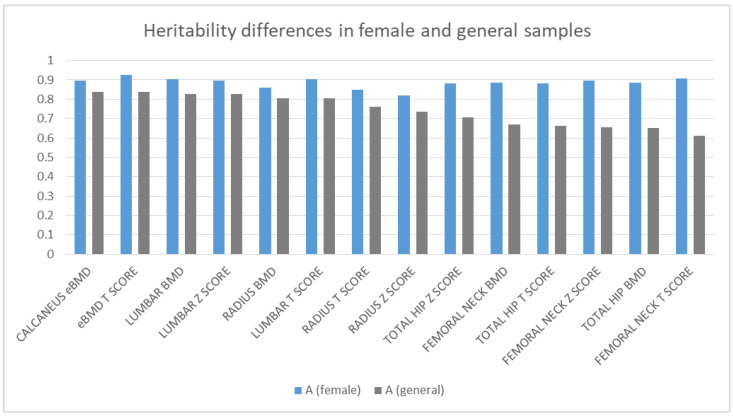
Differences in additive genetic effects in the female and original sample.

**Table 1 medicina-57-00248-t001:** Descriptive statistics.

Variable	Total (*n* = 216)	MZ (*n* = 124)	DZ (*n* = 92)
Sex	72:28	66:34	80:20
Age	54.2 ± 14.3	52.34 * ± 14.48	56.76 * ± 13.72
BMI	25.63 ± 4.72	25.2 ± 4.54	26.2 ± 4.92
LUMBAR BMD	1 ± 0.15	1 ± 0.15	0.99 ± 0.16
LUMBAR Z SCORE	0.35 ± 1.25	0.28 ± 1.23	0.44 ± 1.26
LUMBAR T SCORE	−0.73 ± 1.38	−0.68 ± 1.37	−0.8 ± 1.41
FEMORAL NECK BMD	0.79 ± 0.15	0.8 ± 0.16	0.77 ± 0.13
FEMORAL NECK Z SCORE	0.23 ± 1	0.22 ± 1.06	0.25 ± 0.92
FEMORAL NECK T SCORE	−0.94 ± 1.08	−0.89 ± 1.16	−1 ± 0.99
TOTAL HIP BMD	0.92 ± 0.15	0.94 ± 0.15	0.9 ± 0.14
TOTAL HIP Z SCORE	0.29 ± 1.02	0.31 ± 0.98	0.27 ± 1.07
TOTAL HIP T SCORE	−0.44 ± 1.12	−0.41 ± 1.12	−0.49 ± 1.13
RADIUS BMD	0.65 ± 0.09	0.66 * ± 0.09	0.63 * ± 0.09
RADIUS Z SCORE	−0.39 ± 1.01	−0.4 ± 1.01	−0.38 ± 1.02
RADIUS T SCORE	−1.53 ± 1.18	−1.41 ± 1.14	−1.68 ± 1.22
CALCANEUS eBMD	0.52 ± 0.13	0.53 ± 0.13	0.51 ± 0.12
CALCANEUS eBMD T SCORE	−0.6 ± 1.14	−0.54 ± 1.17	−0.68 ± 1.11

*T*-test: *: *p* < 0.05, n refers to the number of individuals included in an examination. Format is mean ± SD, and percent of female: male for sex. MZ: monozygotic, DZ: dizygotic twins. Variables determined by ultrasound technique.

**Table 2 medicina-57-00248-t002:** ACE models.

Measure	rMZ	rDZ	A	C	E	Model Fit
LUMBAR BMD	0.834 (0.73 0.9)	0.304 (0.007 0.553)	0.828 (0.726, 0.89)	0	0.172 (0.11, 0.274)	1
LUMBAR Z SCORE	0.828 (0.724 0.896)	0.325 (0.036 0.564)	0.828 (0.728, 0.889)	0	0.172 (0.111, 0.272)	1
LUMBAR T SCORE	0.824 (0.695 0.901)	0.291 (−0.044 0.568)	0.806 (0.676, 0.882)	0	0.194 (0.118, 0.324)	0.950
FEMORAL NECK BMD	0.679 (0.507 0.798)	0.066 (−0.236 0.357)	0.669 (0.511, 0.779)	0	0.378 (0.253, 0.554)	1
FEMORAL NECK Z SCORE	0.715 (0.557 0.822)	0.162 (−0.139 0.437)	0.656 (0.492, 0.77)	0	0.331 (0.221, 0.489)	1
FEMORAL NECK T SCORE	0.665 (0.466 0.8)	0.054 (−0.284 0.38)	0.613 (0.409, 0.754)	0	0.387 (0.246, 0.591)	1
TOTAL HIP BMD	0.659 (0.469 0.787)	0.29 2 (−0.007 0.543)	0.653 (0.477, 0.773)	0	0.347 (0.227, 0.523)	1
TOTAL HIP Z SCORE	0.696 (0.53 0.81)	0.318 (0.026 0.56)	0.705 (0.548, 0.809)	0	0.295 (0.191, 0.452)	1
TOTAL HIP T SCORE	0.654 (0.446 0.795)	0.25 (−0.088 0.537)	0.664 (0.462, 0.794)	0	0.336 (0.206, 0.538)	1
RADIUS BMD	0.795 (0.671 0.874)	0.375 (0.086 0.606)	0.806 (0.694, 0.875)	0	0.194 (0.125, 0.306)	1
RADIUS Z SCORE	0.73 (0.58 0.831)	0.452 (0.183 0.658)	0.737 (0.606, 0.825)	0	0.263 (0.175, 0.394)	0.557
RADIUS T SCORE	0.742 (0.575 0.85)	0.391 (0.073 0.637)	0.761 (0.611, 0.853)	0	0.239 (0.147, 0.389)	1
CALCANEUS eBMD ⸸	0.834 (0.721 0.902)	0.564 (0.319 0.739)	0.838 (0.742, 0.896)	0	0.162 (0.104, 0.258)	0.259
CALCANEUS eBMD T SCORE ⸸	0.853 (0.739 0.917)	0.596 (0.335 0.771)	0.838 (0.735, 0.899)	0	0.162 (0.101, 0.265)	0.182

95% confidence interval included in parenthesis. Model fit refers to the *p*-value of the chi-square test of the model, compared to the base ACE model. Here the best models are included, based on Akaike information criteria comparisons. rMZ: intrapair correlation in monozygotic twins, rDZ: intrapair correlation in dizygotic twins, A: additive genetic effects, C: common environment, E: unique environment, ⸸: variables determined by ultrasound technique.

## Data Availability

The data presented in this study are available on request from the corresponding author. The data are not publicly available due to ethical reasons.

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
