# Peer review of "Strong Genetic Effects on Bone Mineral Density in Multiple Locations with Two Different Techniques: Results from a Cross-Sectional Twin Study"

_medicina, 2021, doi:10.3390/medicina57030248_

Round 1
Reviewer 1 Report
Valuable contribution to the literature. I have a series of questions that aim at clarification.
Intro: British classical twin study aiming to determine the genetic component : please rephrase, so that is it clear you refer to the classical twin design which compares MZ to DZ resemblance (e.g. Classical twin studies and beyond. Boomsma D, Busjahn A, Peltonen L. Nat Rev Genet. 2002;3(11):872-82. doi: 10.1038/nrg932).
Could you contrast, either here or in discussion, lifestyle in Hungary and UK? Would you have reason to expect heritability differences given maybe differences in environmental exposure?
The study employed X-ray (DEXA) and heel bone ultrasound technique. Were Z and T scores obtained for both techniques? (what is in e.g. table 1)?
My most important question concerns the model fitting. The correlations for MZ and DZ twins in table 2 very clearly indicate that rMZ > 2*rDZ; this is first evidence for genetic non-additivity. In other words, the authors should fit an ADE model to the data, there is no evidence that C plays a role.
In the GWAS literature, different genes for sites in the body have been identified. Do the authors want to discuss these findings?
They could consider estimating genetic correlations between their outcome traits, to support these GWAS findings.
Reviewer 2 Report
The reviewed manuscript is very interesting. The study design is excellent and the study cohort is unique. However, I am afraid that the authors made two significant methodological errors affecting the presentation of the results and concluding.
First, the authors assumed that the proportion of the influence of genetic and environmental factors on the BMD value is a constant individual feature and made an attempt to calculate the contribution of genetic factors. To test their hypothesis, they used a cohort of subjects at mean age of 54 years. The age range is not given, but I can deduce from the descriptive statistics that study subjects are around between 35 and 75 years. It is highly probable that the proportion of genetic and environmental factors influencing BMD changes throughout our lives - the longer we live, the more the BMD value is modified by environmental factors. Thus, the appropriate cohort to achieve the assumed study aim would be a group of young adults in whom DXA / QUS measurement was performed at the age of peak bone mass (25-30 years). In order to verify whether an attempt to precisely calculate the share of genetic factors in the BMD value in a wide range of middle and older age is still justified, I suggest performing the following additional analysis: calculating the intra-pair BMD (and/or the T-score) difference for each pair of twins and correlating this value with age. If the intra-pair BMD difference does not increase with age, then the way of presenting the results applied by the authors can be maintained. Otherwise, the results presentation should be limited to showing the differences between the MZ and DZ twins.
Second, the authors performed the analysis in a mixed male and female cohort and used the directly measured BMD value as the dependent variable. Such analysis provides misleading results as BMD is a sex-dependent variable and sex has not been included in the independent variables. Interestingly, the authors themselves drew attention to this problem in the discussion, pointing to discrepancies in the results of analyzes based on BMD and T-score (lines 203-210). Indeed, one could expect that analyzes of BMD and T-score provide always coherent results, as the T-score is a simple mathematical transformation of BMD value but not an independent DXA parameter. However, the authors left the observed phenomenon unexplained. The explanation is very simple - if it is not possible to conduct analyzes in the group of men and women separately, then the analysis in the mixed group should be limited to the transformed T-score values only, which ensures sex-adjustment (additionally the Z-score based analysis may be also presented, which provides sex- and age-adjustment). In my opinion, therefore, in the mixed group, all analyzes based directly on BMD should be completely omitted and the results of analyzes concerning only T-score and Z-score should be presented.
When these essential changes to the presentation of the results and appropriate changes in the discussion are made, the paper will be ready for detailed review.
Round 2
Reviewer 2 Report
Thank you for all the explanations and changes made to the manuscript. I find it to be much improved.